# A Comparison of Two Interaction Paradigms for Training Low Cost Automation Assembly in Virtual Environments †

**Federico Manuri** [1,*] , **Federico Decataldo** [1] , **Andrea Sanna** [1] and **Paolo Brizzi** [2]

1    Dipartimento di Automatica e Informatica, Politecnico di Torino, C.so Duca degli Abruzzi 24, 10129 Torino, Italy

2    Competence Industry Manufacturing 4.0 S.C.A R.L, Corso Settembrini 178, 10135 Torino, Italy

*    Correspondence: federico.manuri@polito.it

†    In Proceedings of the IEEE International Conference on Metrology for eXtended Reality, Artificial Intelligence, and Neural Engineering, Rome, Italy, 26–28 October 2022, Entitled Prototyping Industrial Workstation in the Metaverse: A Low Cost Automation Assembly Use Case.

**Abstract:** Virtual environments have been widely adopted for design and training tasks in the industrial domain. Low-cost automation (LCA) is a technology that automatizes some activities using mostly standard automation mechanisms available off the shelf. However, LCA systems should adapt to existing standard production lines and workstations. Thus, workers must customize standard LCA templates and perform adaptation and customization steps. This activity can be very time consuming with physical LCA systems, and in case of errors, it may be necessary to rebuild many parts from scratch. Thus, LCA systems would greatly benefit from a design and prototyping step experienced in a virtual simulation environment. An immersive virtual reality (IVR) application for rapid and easy prototyping of LCA solutions has been investigated in previous work; the assessment of the system usability proved that the users highly appreciated the proposed solutions. This research explores further improvements to exploit the existing IVR application as a training tool for LCA prototyping trainees. The proposed application now provides users with two different interaction paradigms based on the VIVE controllers and the Manus Prime II data gloves. The application's interface has been revised to allow a proper comparison of the two interaction models. The two interfaces have been compared, involving 12 participants in an LCA building task. The System Usability Scale (SUS) and the NASA Task Load Index (TLX) questionnaires have been used to assess the usability and workload of the two solutions.

**Keywords:** virtual reality; virtual prototyping; virtual training; industry 4.0; low-cost automation; virtual manufacturing

## 1. Introduction

Manufacturing companies have undergone profound transformations in recent years, modifying their production strategy towards customized products and fast time to market [1]. This has been possible through a digital revolution fueled by integrating innovative technologies in manufacturing, as depicted by the best practices promoted by Industry 4.0 [2]. Intelligent systems have a decisive impact on cost efficiency, performance levels, quality, and fault-free processes in industries [3]. Among the nine pillars of Industry 4.0, one of the key innovations is represented by the concept of simulation: simulation is usually intended as mathematical modeling that optimizes the process; however, digital tools to design and prototype production processes are the base for optimization activities that fuel time and cost reductions. Virtual simulation environments (VSEs) are a good alternative to physical environments because they are inexpensive, and experiments can be conducted easily and without risks: thus, VSEs have been widely adopted for training activities and prototyping [2].

The diffusion in recent years of virtual reality (VR) headsets, which enable the immersive virtual reality (IVR) paradigm, can provide new levels of interactivity and simplicity for design and prototyping tasks [4]: IVR enables the user to explore the environment virtually while moving in the physical world, evaluating the efficacy of the design choices from different points of view. Moreover, IVR environments enable virtual prototyping of complex workstations and/or industrial production lines, greatly reducing the design cost and time. This high-fidelity virtualization of real industrial scenarios can be used as the perfect training ground to help trainees learn novel technologies or practices. Finally, the ability to share IVR environments with people far away connected through the Internet can enhance remote collaborations among users and one-trainer-multiple-trainee scenarios.

Low-cost automation (LCA) is a technology that automatizes some activities using mostly standard automation mechanisms available off the shelf [5]. An LCA solution is generally characterized by:

- Zero or very little power consumption;
- A few actuators;
- High flexibility;
- High reliability;
- Small dimensions;
- Minimum maintenance;
- Minimum investment/running costs.

An LCA solution usually assures flexibility and improves productivity, providing increased work efficiency and reducing costs. Karakuri automation [6] is the base concept of LCA systems: 'Karakuri' is a Japanese word that means 'achieving motion with no power or low power', referring to an ancient Japanese technique to move dolls without actuators or power. This technique is used to achieve unique motion or to convert one form of motion into another (e.g., vertical to horizontal and linear to rotary) using different mechanisms, such as cam and follower, counterweights, and rope and pulley. LCA solutions have been widely used to reduce manual operations in highly automated systems and production lines [7]. However, LCA systems should adapt to existing standard production lines and workstations. Thus, workers must customize standard LCA templates and perform adaptation and customization steps. This activity can be very time consuming with physical LCA systems, and in case of errors, it may be necessary to rebuild many parts from scratch. Thus, LCA systems would greatly benefit from a design and prototyping step experienced in a VSE.

This research explores further improvements to exploit the immersive virtual environment (IVE) developed in [8] as a training tool for LCA prototyping trainees. The challenges of assembling an LCA solution include direct hand interaction and manipulation with multiple objects, which need to be assembled together but allow almost infinite possible combinations, thus requiring high-precision tracking for the hands' position. This may lead to multiple configurations with hands occlusion or limited fingers visibility, thus greatly affecting the system's robustness of vision-based systems for real-time hand tracking. Moreover, since the use case requires direct hand interaction, indirect object manipulation with gestures has not been considered. Thus, the existing application has been extended with and hand-tracking interaction paradigm and compared to the controller-based one. To this end, the interaction paradigm is the only variable parameter; all the other parameters that may affect the user experience in IVR applications are fixed. The scope of this work is identified by the following research questions:

1. R1 : is the hand-tracking interaction paradigm more usable than the controller-based paradigm?
2. R2: is the level of workload involved in the hand-tracking interaction paradigm lower than that of the controller-based paradigm?

Based on the users' feedback, both interfaces provided users with a good level of satisfaction and usability, with a preference for the Manus gloves solution. The question-

naire results do not show statistically significant differences between the two interfaces. However, these preliminary tests provided the data to compute a priori power analysis to establish the smallest sample size required to measure a statistically meaningful effect size.

This paper is organized as follows: Section 2 provides an overview of the state of the art and the contribution of the proposed paper. Section 3 describes the proposed system and the two interaction paradigms developed for the LCA assembly tasks. Section 4 describes the test carried out to compare the two interaction models and the analysis of the results.

## 2. State of the Art

### 2.1. Virtual Environments

Virtual reality has been investigated as a relevant technology to design and prototype assembly tasks since the 1990s [9]. The virtual assembly design environment (VADE) [10] resulted from a research project that started in 1995 to explore the potential and technical challenges of using VR technologies for design and manufacturing. The VADE system was one of the first virtual reality environments developed for evaluation tasks and assembly planning. Meanwhile, Dewar et al. [11] focused on developing novel tools to simplify the interaction in the virtual environment, exploring the usage of VR for assembly tasks. More specifically, the authors investigated novel approaches to assist the assembly planner in joining two objects in a virtual world, proposing proximity and collision snapping tools. After a decade of research, the main challenges for virtual assembly tasks are still inter-part constraint detection, collision detection, and management of physics-based modeling, as depicted in a review of VR for assembly methods prototyping [12]. Liu et al. [13] researched novel methods to overcome the limitations of constraint-based virtual assembly methods, focusing on physics-based virtual assembly techniques. Francillette et al. [14] developed an open-source simulator to build virtual intelligent environments, providing a set of virtual sensors reproducing the behavior of the real ones, the capability to simulate interactions, and gathering data. Noghabaei et al. [15] proposed a virtual manipulation platform for training workers building objects in an IVE wearing a VR headset. Workers can assemble multiple virtual modules, identifying potential problems and discrepancies of as-built elements. Sharpa et al. [16] demonstrated both assembly and 3D presentation for an aircraft in a collaborative virtual assembly environment. Pan et al. [17] developed a virtual simulation assembly system of robotic parts in an IVE for training activities. The scope of the system was to enhance students' capabilities in assembling robotic parts using interactive, virtual equipment. Halabi [18] investigated the effectiveness of VR with a project-based learning approach in engineering design projects, proving that VR can improve learning outcomes and promote effective communication. Shamsuzzoha et al. [19] evaluated a digital factory-based VR platform for an industrial training and maintenance system. Results suggest that using VR platforms to train and maintain complex industrial tasks should be encouraged. Overall, VSEs based on IVR represent a compelling tool for training activities, collaborative prototyping, and evaluating and planning product assembly.

### 2.2. Hand-Tracking Interfaces

Virtual reality headsets usually provide users with physical controllers to interact with a virtual 3D environment effectively. However, hand-tracking interfaces have been investigated to assess if they can increase the sense of immersion, embodiment, and usability compared to standard controllers. In [20], 22 participants performed a 3D manipulation task (opening a virtual door) with a custom cyber-glove system and the HTC VIVE controller. Results showed that cyber-glove-based interactions performed equal to or better than controller-based ones objectively and subjectively, whereas 90% of users experienced an equal or increased sense of embodiment. Khundam et al. [21] assigned a VR intubation training case study to 30 medical students to investigate the differences between using controllers and hand tracking in medical interactions. The results showed that the interaction time, satisfaction, and usability were similar. However, medical training with free-hand gestures is more natural for real-world situations. Moreover, hand-tracking solutions allow

trainees to recognize and correct their postures intuitively, and thus, should be beneficial for self-learning and practicing. Rantamaa et al. [22] investigated interaction methods in VR for object manipulation and marking tasks in medical surgery planning, comparing mouse, hand-tracking, and a combination of a VR stylus and a grab-enabled VR controller. Results showed that the combination of stylus and controller was the preferred interaction method because it was considered more similar to using bare hands compared to the others. In contrast, the mouse interaction method was objectively the most accurate. Overall, these recent studies show different results in user preferences towards controller-based or gesture-based solutions; moreover, results are highly dependent on the specific use case.

### 2.3. Contribution

An immersive virtual reality application for rapid and easy prototyping of LCA solutions has been investigated in a previous work [8], and the assessment of the system usability proved that it was highly appreciated by the users. Moreover, due to the difficulties and limitations of LCA prototyping in the real world, an IVE could be useful as a training tool for workers with little to no experience with LCA systems. To this end, it is relevant to investigate further the usage of data gloves instead of physical controllers to provide easier interactions in the virtual world and to let the user focus on the task at hand. This research explores further improvements to exploit the IVE developed in [8] as a training tool for LCA prototyping trainees. The application has been extended to include the Manus Prime II data gloves as an alternative to the interaction paradigm based on the VIVE controllers [8]. The application's interface has been revised to allow a proper comparison of the two interaction models. The two interfaces have been compared, involving 12 participants in the same LCA building task defined in [8]. The SUS and the NASA-TLX questionnaires have been used to assess the usability and workload of the two solutions.

### 3. The Immersive Virtual Reality Prototyping Environment

The proposed research aims to simplify the training process of LCA creators by using an IVE for LCA prototyping, comparing two different interaction paradigms to evaluate the easiest solution for trainees. Figure 1 shows the architecture of the system. To provide a different interaction paradigm in the IVE, the Manus Prime II data gloves have been integrated into the solution presented in [8]. These data gloves track individual fingers on two joints, supporting 11 degrees of freedom on every finger. The Prime II gloves are an easily set-up, portable solution to produce high-fidelity finger tracking with a short, 45-s calibration process and a latency under five milliseconds, ideal for simulation and virtual training. The Manus Software Development Kit (SDK) 2.1 and plugin simplify the data streaming in the Unity 3D game engine, providing realistic hand and finger motions in VR experiences. Moreover, the gloves can be used with any SteamVR-compatible headset, including the HTC VIVE Pro. Thus, the IVE application developed in Unity 3D has been upgraded to be compatible with the Manus Prime II data gloves. The Manus Core application handles the data stream from the gloves; it comes with an SDK that enables data stream integration into any application and can stream live data in real time to different native plugins. The Manus Dashboard is an intuitive, user-friendly interface for easy managing, calibrating, and recording. It quickly and accurately calibrates the Manus gloves with three simple gestures. Furthermore, it enables users to define the Virtual Reality Hardware Setup by exchanging data with the SteamVR application. To track the hand position in the 3D space, each Manus glove was paired with a VIVE tracker 2.0 through the graphical interface of the Manus Dashboard. These trackers create a wireless and seamless connection between the attached, real-world tool and the VIVE system, making tracking any physical object in the virtual environment possible.

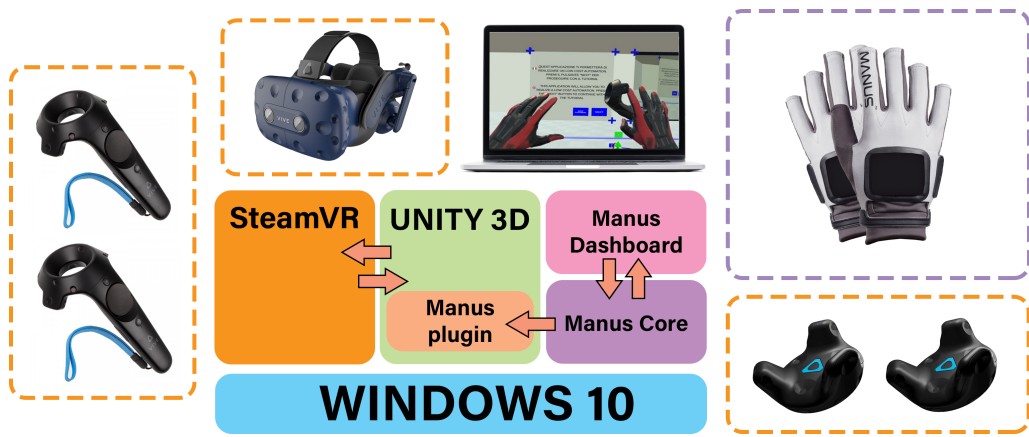

**Figure 1.** The architecture of the system. The devices circled in orange exchange data with SteamVR, whereas the Manus gloves exchange data with the Manus Core application.

The combination of the Manus gloves and the VIVE trackers provides the alternative to the VIVE controllers interaction system: the Manus gloves provide the data defining the hand poses and gestures, which determines the user intention. The VIVE trackers provide the positional data necessary to understand the context, defining the virtual object involved in the interaction process based on their proximity with the virtual hands. The Manus Unity Plugin conveys all these data to the Unity 3D application.

The primary interaction paradigm adopted in the proposed system relies on the user's hand positions in the virtual space. The actions available to the user depend on the system's status, which relies on the concept of collision: the Unity physic simulation engine identifies starting, ending, or ongoing collisions between objects in the scene.

The system provides the following functionalities:

- Virtual menus: these menus provide both information and buttons, which can be virtually pressed to activate different options, e.g., adding 3D objects to the scene;
- Grab & release: used to select and move objects in the 3D scene;
- Connecting objects: used to create complex LCA by connecting simple objects by colliding them; connections depend on the concepts of compatibility and connections points, which have been explored in [8];
- Separating objects: used to remove an object or objects group from a complex object;
- Scaling objects: used to change one or more dimensions of a simple object, e.g., the length of a pipe.
- Adjusting objects: used to change an object or an object group position without detaching it from a complex object, e.g., to adjust the vertical position of a pipe;
- Copy position: used to copy one of the three object's coordinates to or from another one;
- Copy size: used to copy the object size from another one of the same type;
- Split: used to split a complex object into its parts;
- Distance: used to compute the gap between the current object and another.

These actions are made available to users through direct actions (physical buttons on the VIVE controller or hand poses when using the manus gloves) or menus.

The actions mapped on the VIVE controller buttons are the following:

- Menu button: used for menu management. When pressed, the menus available in the application are opened or closed according to the system's status;
- Trigger button: this is the button on the back of the controller; when pressed, it starts the actions of grabbing an object; when released, the grabbed object is detached from the hand.

The first version of the proposed system mapped two other functionalities on available physical buttons of the VIVE controller, more specifically:

- Grip button: this is the button on the sides of the controller; when pressed, if pointing at an object made up of a complex structure, it allows you to detach it from the structure;
- Touchpad: this is the central button on the front of the controller; when pressed together with the trigger, if pointing at a compound object within a complex structure, it allows you to change its position without detaching it, thus facilitating fine pose correction.

However, in the current application version, these functionalities are mapped on the virtual buttons available through the contextual menus. This change is based on two principles.

The first one is the easiness of use: preliminary tests with the VIVE controllers showed that the grip button and touchpad were not easily associated with their related functionalities. In contrast, the trigger and menu buttons were easily recognizable by the users and overall improved the learnability effect of the corresponding functionalities.

The second one is the principle of correspondence, which means that the two interfaces should provide the same approach, either direct actions or menu voices, to make available to the user the same functionalities. Thus, since other functionalities were already made available through menus, removing these mappings and providing the detach and adjust functionalities through menu options was logical. Finally, the user's interaction with the buttons displayed on the virtual panels relies on the VIVE controller collision with virtual buttons.

The same actions are mapped on different hand poses, as shown in Figure 2:

- Open hand: detected when the user completely opens one hand. Used to release an object after a grab action, or to show the main menu panel, if the hand is facing the user.
- Grab pose: detected when the user completely closes one hand. If the hand collides with an object, the grab action starts.
- Pointing pose: used to select different options in the virtual menus, or to open contextual menus by touching objects in the 3D scene.

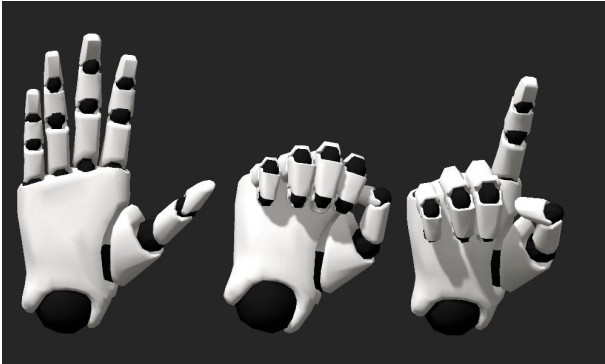

**Figure 2.** The three different hands poses tracked by the system: open hand (**left**), grab pose (**center**), and pointing pose (**right**).

Finally, the scaling object function is provided the same way for both the VIVE controllers and the Manus gloves, by grabbing an object with both hands and stretching or squeezing it.

## 4. Test and Results

The LCA building task defined in [8] has been proposed to the testers to compare the usability of the two proposed interfaces. Twelve users participated in the tests, nine males and three females, aged between 19 and 25. Before the experiment, they signed a consent form explaining the study's goals and the anonymity of the subject's data. Each user performed the tasks using both interfaces, with half of the participants starting with the gloves interface and the other half with the controllers to avoid learnability effects.

*4.1. Test Procedures*

The overall procedure can be summarized as follows:

1.  Introduction: the user is introduced to the proposed research, and the test procedure is explained in detail;
2.  User data: the user fills out a preliminary form to collect general user information and to ask some questions pertaining to their previous experiences with VR;
3.  Hardware setup: the user is equipped with the VIVE HTC Pro headset; if the Manus gloves are used, the calibration procedure is carried out;
4.  Virtual environment tutorial: when the application starts, after selecting the desired interaction interface, a set of instruction panels guides the user, showing the application functionalities; each panel is composed of a textual description, images, and/or videos displaying the application usage; if the controllers are displayed in the panel, interacting with the different buttons of the physical controller highlights its graphical counterpart on the virtual panel. The panels guide the user into practicing with the available commands, e.g., adding objects to the scene, grabbing, connecting, or scaling them. When the tutorial ends, testers are given 5 more minutes to explore the virtual environment further and to try out the available commands and menus;
5.  LCA assembly task: after the tutorial step, the users are asked to assemble an LCA starting from an existing reference displayed in the virtual environment. The 3D reference is not only a graphical representation of the target LCA; it was assembled following the virtual environment mechanism. Thus, it is possible to measure the distances between different parts of the reference LCA or to copy the size of its parts;
6.  Questionnaires: the user had to complete the SUS [23] and the NASA-TLX [24] questionnaires;
7.  The user repeated steps 5–6 using the other interface;
8.  Interview: the user is then interviewed regarding the overall experience.

*4.2. Measurements*

Testers were asked to complete two subjective questionnaires to evaluate the two proposed interfaces: the System Usability Scale questionnaire, which is widely adopted to measure a system's usability and to compare systems, and the NASA Task Load Index questionnaire, which is used to assess the workload of a given task. The time spent completing the LCA assembly task was measured for each user and for both interfaces. Errors in the assembly task have been counted when a user needs to delete something from the scene because it is useless or has to be rebuilt from scratch, and questions to the trainer when the trainee was not able to move on with the task have been counted as well.

*4.3. Results*

Based on the information provided in the preliminary form, ten out of twelve users had previous experiences with VR. Table 1 shows the SUS scores, completion times, errors, and questions for the assembly task performed with the VIVE controllers and the Manus gloves. Table 2 shows the NASA-TLX weighted workload scores on a 0–100 scale for the VIVE controllers and the Manus gloves. Table 3 shows the NASA-TLX weighted individual scores for the VIVE controllers, whereas Table 4 shows the NASA-TLX weighted individual scores for the Manus gloves. Table 5 shows the NASA-TLX raw individual mean scores on a 0–100 scale for the VIVE controllers and the Manus gloves.

**Table 1.** SUS scores, completion times (in seconds), errors, and questions for the assembly task performed with the VIVE controllers and the Manus gloves.

| User | SUS Scores | | Completion Times | | Errors | | Questions | |
|---|---|---|---|---|---|---|---|---|
| | VIVE | Manus | VIVE | Manus | VIVE | Manus | VIVE | Manus |
| 1 | 85 | 77.5 | 1378 | 772 | 2 | 2 | 3 | 1 |
| 2 | 35 | 75 | 828 | 754 | 1 | 0 | 1 | 2 |
| 3 | 75 | 80 | 522 | 400 | 0 | 1 | 0 | 1 |
| 4 | 57.5 | 82.5 | 530 | 360 | 0 | 1 | 0 | 1 |
| 5 | 67.5 | 60 | 866 | 736 | 3 | 3 | 2 | 1 |
| 6 | 65 | 80 | 400 | 702 | 0 | 0 | 1 | 3 |
| 7 | 70 | 90 | 1674 | 883 | 1 | 1 | 4 | 1 |
| 8 | 60 | 67.5 | 707 | 743 | 1 | 2 | 2 | 2 |
| 9 | 87.5 | 87.5 | 486 | 545 | 1 | 0 | 1 | 0 |
| 10 | 80 | 80 | 831 | 601 | 0 | 1 | 2 | 0 |
| 11 | 90 | 87.5 | 366 | 307 | 0 | 1 | 0 | 0 |
| 12 | 57.5 | 75 | 609 | 571 | 1 | 1 | 2 | 0 |
| **Average** | **69.17** | **78.54** | **766.42** | **614.5** | **0.84** | **1.08** | **1.5** | **1** |

**Table 2.** NASA-TLX weighted workload scores on a 0–100 scale for the VIVE controllers and the Manus gloves.

| | NASA-TLX Weighted Total Work Load | |
|---|---|---|
| **User** | **VIVE** | **MANUS** |
| 1 | 49.33 | 42.00 |
| 2 | 89.33 | 45.00 |
| 3 | 40.67 | 51.33 |
| 4 | 43.33 | 24.67 |
| 5 | 58.00 | 46.67 |
| 6 | 36.67 | 19.33 |
| 7 | 44.00 | 18.00 |
| 8 | 47.33 | 38.00 |
| 9 | 24.67 | 37.33 |
| 10 | 48.00 | 28.67 |
| 11 | 52.67 | 55.33 |
| 12 | 48.67 | 42.67 |
| **Mean** | **48.56** | **37.42** |

**Table 3.** NASA-TLX weighted individual scores for the VIVE controllers.

| | NASA-TLX Weighted Subscores—VIVE Controllers | | | | | |
|---|---|---|---|---|---|---|
| **User** | **Mental** | **Physical** | **Temporal** | **Performance** | **Effort** | **Frustration** |
| 1 | 120 | 0 | 100 | 300 | 200 | 20 |
| 2 | 360 | 0 | 400 | 180 | 100 | 300 |
| 3 | 300 | 50 | 90 | 80 | 80 | 10 |
| 4 | 160 | 0 | 200 | 100 | 40 | 150 |
| 5 | 210 | 20 | 200 | 80 | 300 | 60 |
| 6 | 160 | 20 | 10 | 300 | 60 | 0 |
| 7 | 240 | 0 | 100 | 150 | 50 | 120 |
| 8 | 150 | 0 | 80 | 250 | 80 | 150 |
| 9 | 160 | 0 | 60 | 100 | 40 | 10 |
| 10 | 120 | 0 | 10 | 240 | 200 | 150 |
| 11 | 210 | 40 | 150 | 150 | 240 | 0 |
| 12 | 100 | 0 | 350 | 120 | 40 | 120 |

**Table 4.** NASA-TLX weighted individual scores for the Manus gloves.

| | NASA-TLX Weighted Subscores—Manus Gloves | | | | | |
|---|---|---|---|---|---|---|
| **User** | **Mental** | **Physical** | **Temporal** | **Performance** | **Effort** | **Frustration** |
| 1 | 240 | 0 | 120 | 150 | 60 | 60 |
| 2 | 200 | 20 | 200 | 200 | 100 | 0 |
| 3 | 350 | 20 | 240 | 90 | 30 | 40 |
| 4 | 60 | 0 | 80 | 120 | 20 | 90 |
| 5 | 160 | 120 | 20 | 280 | 120 | 0 |
| 6 | 80 | 20 | 20 | 150 | 20 | 0 |
| 7 | 90 | 10 | 50 | 80 | 0 | 40 |
| 8 | 100 | 0 | 150 | 200 | 80 | 40 |
| 9 | 200 | 0 | 80 | 100 | 80 | 100 |
| 10 | 150 | 40 | 10 | 150 | 80 | 0 |
| 11 | 240 | 140 | 210 | 100 | 140 | 0 |
| 12 | 150 | 20 | 240 | 150 | 80 | 0 |

**Table 5.** NASA-TLX raw individual mean scores on a 0-100 scale for the VIVE controllers and the Manus gloves.

| NASA-TLX Raw Individual Mean Scores | | |
|---|---|---|
| | **VIVE** | **MANUS** |
| Mental demand | 55.83 | 47.50 |
| Physical demand | 29.17 | 24.17 |
| Temporal demand | 45.83 | 34.17 |
| Performance | 48.33 | 34.17 |
| Effort | 52.50 | 36.67 |
| Frustration | 35.83 | 26.67 |

*4.4. Discussion*

The following procedure has been used to analyze the collected data:

1. Normality data assessment by the Shapiro–Wilk test;
2. Case Data Normally Distributed: dependent *t*-test;
3. Case Data Non-Normally Distributed: if the distribution of differences (DDs) is symmetrical, Wilcoxon signed-rank test. Otherwise, sign test.

The SUS data for both the VIVE controllers and the Manus gloves present normal distributions ($p_{VIVE} = 0.59$, $p_{MANUS} = 0.42$) and the dependent *t*-test does not show statistically significant differences between the two interfaces ($p = 0.083$). The NASA-TLX data for the VIVE controllers are not normally distributed ($p_{VIVE} = 0.029$, $p_{MANUS} = 0.54$). Since the dependent *t*-test should be used only if both groups present uniform distributions, all NASA-TLX results have been considered as not normally distributed. DDs are not symmetrical, and the sign test shows no statistically significant differences ($p = 0.08326$). Thus, it is not possible to reject the null hypothesis for R1 and R2.

Based on the average SUS scores, both interfaces obtained a score above 68, which is the minimum to consider a system sufficiently usable. However, the gloves were more appreciated than the controllers, with an average SUS score of 78.54, which represents a usability value between good and excellent, according to the rating scale provided by Bangor et al. [25]. Due to the limited number of tests, it is not possible to statistically prove that the gloves are more usable than the controllers. However, the data obtained with these preliminary tests can be used for a priori power analysis to compute the smallest sample size required to measure a statistically meaningful effect size. Based on the SUS scores, means, and standard deviations, the expected effect size is $d = 0.74$. With an 80% chance of detecting the effect if it is true (20% of a Type II error) and a 5% chance of detecting an effect if there is no such effect (Type I error), a sample size of 31 participants for each group would be necessary.

The NASA-TLX results show that the task is perceived as less demanding when the Manus gloves are used to interact in the virtual environment. Individual raw mean scores confirm these results; the most demanding factors are mental demand and effort, confirming the test carried out in the previous work.

Error averages show that the user committed almost the same number of errors with both interfaces, with an average slightly in favor of the VIVE controllers; on the other hand, users had to ask more questions when using the VIVE controllers interface, suggesting that the Manus gloves interface is more straightforward in its usage.

Based on the feedback provided by the users in the interviews, most of them were satisfied with their performances in accomplishing the proposed task. This result is confirmed by the individual NASA-TLX scores, since both the frustration and the performance values are quite low, corresponding to a good level of perceived satisfaction. The users were also asked to suggest improvements or new features to the existing system. Some of them proposed adding a calibration panel for the Manus gloves inside the application to tune the threshold values of each finger used to identify the different hand poses. Another request is to make the interaction with the tutorial panels and menus more similar to the interaction with the LCA parts, adding the grabbing functionality to move them around in the 3D environments instead of spawning them at a predefined position based on the user field of view.

### 5. Conclusions and Future Works

Manufacturing companies have undergone profound transformations in recent years, and digital simulations of production processes are the base for optimization activities through novel technologies, fueling time and cost reductions. Virtual simulation environments have been widely adopted for training activities and prototyping, exploiting the possibilities offered by cyber-physical systems. Virtual simulators are a good alternative to physical environments because they are inexpensive, and experiments can be conducted easily and safely. The proposed research explores further improvements to exploit the existing IVE proposed in [8] as a training tool for LCA prototyping trainees. The Unity 3D application has been extended to include the Manus Prime II data gloves as an alternative to the VIVE controllers. The application's interface has been revised to allow users a proper comparison of the two interaction paradigms. The two interfaces have been compared, involving 12 participants in an LCA building task. The System Usability Scale and NASA-TLX questionnaires have been used to assess the usability and workload of the two solutions, respectively: both interfaces provided users with a good level of satisfaction and usability, with a preference for the Manus gloves solution. Based on the feedback provided by the users in the interviews, most of them were satisfied with their performances in accomplishing the proposed task. The NASA-TLX scores confirm this result, since the task is perceived as less demanding when the Manus gloves are used to interact in the virtual environment. Moreover, these preliminary tests provided the data to compute a priori power analysis to establish the smallest sample size required to measure a statistically meaningful effect size. Future works will be aimed at developing a version of the proposed prototyping system for a cave automatic virtual environment (CAVE). Since a CAVE does not disconnect the user from the real world, a comparison among different visual experiences will be possible, applying recent assessment methodologies [26,27]. Moreover, finger-tracking gloves with haptic feedback could be investigated, as they may enhance the user's feedback and improve the whole assembly process, contributing to further reducing the task workload.

**Author Contributions:** Conceptualization, A.S., F.M. and P.B.; methodology, A.S., F.M. and P.B.; system architecture, A.S., F.D. and F.M.; design and realization, F.D. and F.M.; software and user interface, A.S. and F.D.; validation and usability analysis, A.S., F.D. and F.M.; investigation and result analysis, F.M. and P.B.; resources, A.S. and F.M.; writing—original draft preparation, A.S. and F.M.; writing—review and editing, P.B. All authors have read and agreed to the published version of the manuscript.

**Funding:** This research received no external funding.

**Institutional Review Board Statement:** Not applicable.

**Informed Consent Statement:** No informed consent required; questionnaire data gathered anonymously with no personal characteristics.

**Data Availability Statement:** All the data collected during this study are available at the following link http://sanna.polito.it/LowCostAutomation/data.csv (accessed on 19 April 2023).

**Conflicts of Interest:** The authors declare no conflict of interest.

## Abbreviations

The following abbreviations are used in this manuscript:

| | |
|---|---|
| CAVE | Cave Automatic Virtual Environment |
| DD | Distribution of Differences |
| IVE | Immersive Virtual Environment |
| IVR | Immersive Virtual Reality |
| LCA | Low-Cost Automation |
| SDK | Software Development Kit |
| SUS | System Usability Scale |
| TLX | Task Load Index |
| VADE | Virtual Assembly Design Environment |
| VR | Virtual Reality |
| VSE | Virtual Simulation Environment |

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
