# Peer review of "A Comparison of Two Interaction Paradigms for Training Low Cost Automation Assembly in Virtual Environments†"

_information, doi:10.3390/info14060340_

Round 1

Reviewer 1 Report

An immersive virtual reality (IVR) app for rapid and easy prototyping of Low-cost Automation is investigated in this paper with a detailed assessment of the system usability. The authors also explored the improvements to exploit the IVR application as a training tool for LCA.  A comparative assessment is presented with the prior literature.

There exist numerous literature on AR assistance in assembly and disassembly, also recommended to see some recent literature with the keywords “Challenges on AR/VR technologies for manufacturing systems”

Hong, Yu, et al. "A marker-less assembly stage recognition method based on corner feature." Advanced Engineering Informatics 56 (2023): 101950.

Eswaran, M., et al. " Augmented reality-based guidance in product assembly and maintenance/repair perspective: A state of the art review." Expert Systems with Applications (2022): 118983.

Maffei, Antonio, et al. "Dynamic Mixed Reality Assembly Guidance Using Optical Recognition Methods." Applied Sciences 13.3 (2023): 1760.

As stated in the abstract, this paper is more on implementation level.

No technical improvements are identified compared to the existing literature, the authors should justify the novelty.

Please give the full form of the abbreviations at their first appearance. (or) more the abbreviations section next to the abstract.

Please list the assumptions followed 

Author Response

Dear Reviewer,

We appreciate your time and effort in carefully evaluating our manuscript titled “A comparison of two interaction paradigms for training Low Cost Automation assembly in virtual environments”. We thank you for your valuable comments and suggestions.

We have carefully considered the Reviewer's comments and recommendations and have updated the paper accordingly. We believe that the changes we have made to the manuscript have significantly improved the clarity and quality of our work. We hope these changes satisfy the Reviewers and that our manuscript can be accepted. An updated version of the proposed paper with all the changes highlighted has been provided, whereas you can find the Reviewer’s requests (in italics) and our answers (in plain text) here:

Request 1. There exist numerous literature on AR assistance in assembly and disassembly, also recommended to see some recent literature with the keywords “Challenges on AR/VR technologies for manufacturing systems”

Hong, Yu, et al. "A marker-less assembly stage recognition method based on corner feature." Advanced Engineering Informatics 56 (2023): 101950.

Eswaran, M., et al. " Augmented reality-based guidance in product assembly and maintenance/repair perspective: A state of the art review." Expert Systems with Applications (2022): 118983.

Maffei, Antonio, et al. "Dynamic Mixed Reality Assembly Guidance Using Optical Recognition Methods." Applied Sciences 13.3 (2023): 1760.

Answer 1. We agree with the reviewer that exists numerous literature on AR assistance in assembly and disassembly. However, the role of AR assembly is to support users interacting with real objects; thus, the main challenges revolve around tracking the real environment, tracking the real objects, the accuracy in aligning digital content to the physical world, and the best way to convey instructions and suggestions without incurring in context-switch problematics. The proposed research instead explores further improvements to exploit an existing Immersive Virtual Reality application as a training tool for Low Cost Automation (LCA) prototyping trainees. LCA systems should adapt to existing standard production lines and workstations. Thus, workers must customize standard LCA templates and perform adaptation and customization steps. This activity can be very time-consuming on physical LCA systems, and in case of errors, it may be necessary to rebuild many parts from scratch. Thus, LCA systems would greatly benefit from a design and prototyping step experienced in a Virtual Simulation Environment.

Request 2. No technical improvements are identified compared to the existing literature, the authors should justify the novelty.

Answer 2. To the best of our knowledge, no previous research compares physical controllers and finger-tracking gloves in an industrial use case similar to LCA prototyping. The challenges of assembling an LCA solution include direct hand interaction and manipulation with multiple objects, which need to be assembled together but allow almost infinite possible combinations: this may lead to multiple configurations with hands occlusion or limited fingers visibility, thus greatly affecting the system robustness of vision-based systems for real-time hand-tracking. Moreover, indirect manipulation of the objects would negatively affect the overall experience; thus, we didn’t consider gestures either.

The following research questions identify the scope of the proposed research:

R1: is the hand-tracking interaction paradigm more usable than the controller-based one?

R2: is the level of workload involved in the hand-tracking interaction paradigm lower than that of the controller-based one?

We aim to determine the best interaction paradigm for the proposed case study. We improved the introduction to explain these concepts better.

Request 3. Please give the full form of the abbreviations at their first appearance. (or) more the abbreviations section next to the abstract.

Answer 3. The whole paper has been revised to address this issue, providing the full form of the abbreviations at their first appearance. The MDPI template defines the position of the abbreviations section and cannot be changed.

Request 4. Please list the assumptions followed 

Answer 4. Our starting assumption is that LCA systems would greatly benefit from a design and prototyping step experienced in a Virtual Simulation Environment, as discussed in our previously published research:

Manuri, F., Gravina, N., Sanna, A., Brizzi, P. (2022, October). Prototyping industrial workstation in the Metaverse: a Low Cost Automation assembly use case. In 2022 IEEE International Conference on Metrology for Extended Reality, Artificial Intelligence and Neural Engineering (MetroXRAINE) (pp. 133-138). IEEE.

Our conclusions resulting from the proposed research are detailed in the “Discussion” section as follows:

  1. Based on the average SUS scores, both interfaces obtained a score above 68, which is the minimum to consider a system sufficiently usable. However, the gloves were more appreciated than the controllers, with an average SUS score of 78, 54, which represents a usability value between good and excellent.
  2. The data obtained with these preliminary tests can be used for a priori power analysis to compute the smallest sample size required to measure a statistically meaningful effect size.
  3. The NASA-TLX results show that the task is perceived as less demanding when the Manus gloves are used to interact in the virtual environment. Individual raw mean scores confirm these results; the most demanding factors are mental demand and effort, confirming the test carried out in the previous work.
  4. Based on the feedback provided by the users in the interviews, most of them were satisfied with their performances in accomplishing the proposed task. This result is confirmed by the individual NASA-TLX scores since both the frustration and the performance values are quite low, corresponding to a good level of perceived satisfaction.

Conclusions #1 and #2 are further summarized in Section 5 (Conclusions and Future Works) as follows:

“Based on the users’ feedback, both interfaces provided users with a good level of satisfaction and usability, with a preference for the Manus gloves solution. Moreover, these preliminary tests provided the data to compute a priori power analysis to establish the smallest sample size required to measure a statistically meaningful effect size.”

We improved Section 5 to summarize conclusions #3 and #4 too.

Reviewer 2 Report

This paper proposes a comparison of two interaction paradigms for training LCA assembly in virtual environments, with 12 participants in an LCA building task. I have several comments as follows:

1. The reasons for only comparing the two paradigms shall be explained more.

2. The main contributions of this paper can be summarized at the end of the introduction part.

3. Except for the interaction, the visual experience of VR environments is significant. Some relevant references are recommended to be reviewed, including No-reference quality assessment for 360-degree images by analysis of multifrequency information and local-global naturalness, Deep virtual reality image quality assessment with human perception guider for omnidirectional image, etc.

4. It is suggested to further improve the presentation. For example, the comma in the tables is confusing.

n/a

Author Response

Dear Reviewer,

We appreciate your time and effort in carefully evaluating our manuscript titled “A comparison of two interaction paradigms for training Low Cost Automation assembly in virtual environments”. We thank you for your valuable comments and suggestions.

We have carefully considered the Reviewer's comments and recommendations and have updated the paper accordingly. We believe that the changes we have made to the manuscript have significantly improved the clarity and quality of our work. We hope these changes satisfy the Reviewers and that our manuscript can be accepted. An updated version of the proposed paper with all the changes highlighted has been provided, whereas you can find the Reviewer’s requests (in italics) and our answers (in plain text) here:

Request 1. The reasons for only comparing the two paradigms shall be explained more.

Answer 1. We agree with the reviewer. We are aware that multiple alternatives could be possible for the interaction. However, the challenges of assembling an LCA solution include direct hand interaction and manipulation with multiple objects, which need to be assembled together but which allow almost infinite possible combinations: this may lead to multiple configurations with hands occlusion or limited fingers visibility, thus greatly affecting the system robustness of vision-based systems for real-time hand-tracking. Moreover, indirect manipulation of the objects would negatively affect the overall experience; thus, we didn’t consider gestures either.

We improved the introduction to explain these concepts better.

On the other hand, finger-tracking gloves with haptic feedback will only enhance the user’s feedback without necessarily improving the assembly process in terms of precision compared to a controller-based solution. However, we may investigate this paradigm in future works.

We improved the future works accordingly.

Request 2. The main contributions of this paper can be summarized at the end of the introduction part.

Answer 2. The main contributions of this paper are summarized at the end of the introduction (page 2, lines 68-90), before the paper organization paragraph, and have been improved accordingly to the other reviewers’ requests.

Request 3. Except for the interaction, the visual experience of VR environments is significant. Some relevant references are recommended to be reviewed, including No-reference quality assessment for 360-degree images by analysis of multifrequency information and local-global naturalness, Deep virtual reality image quality assessment with human perception guider for omnidirectional image, etc.

Answer 3. We agree with the reviewer, the proposed research focuses on the interaction paradigm and does not investigate the visual experience. Future works will be aimed at developing a version of the proposed prototyping system for a Cave Automatic Virtual Environment (CAVE), which offers a different visual experience. The system usability of the proposed application with the controller interaction paradigm has already been evaluated positively in our previously published research:

Manuri, F., Gravina, N., Sanna, A., Brizzi, P. (2022, October). Prototyping industrial workstation in the Metaverse: a Low Cost Automation assembly use case. In 2022 IEEE International Conference on Metrology for Extended Reality, Artificial Intelligence and Neural Engineering (MetroXRAINE) (pp. 133-138). IEEE.

Thus, we assume that the visual experience provided by the proposed application is adequate to perform the proposed comparison.

We improved the introduction to clarify that the user experience in Immersive Virtual Reality applications relies on multiple parameters. Still, in our research, we assume that the only variable parameter is the interaction paradigm, and all the other parameters are fixed.

We improved the future works to clarify that developing a CAVE solution would allow us to investigate the visual experience of the proposed application as well as compare the CAVE and Virtual Reality Headset experiences.

Request 4. It is suggested to further improve the presentation. For example, the comma in the tables is confusing.

Answer 4. The whole paper has been revised to address this issue.

Round 2

Reviewer 2 Report

The authors have addressed most of my comments. One minor point: for the future work of visual experience, the provided reference can be added for better clarity.

n/a

Author Response

Dear Reviewer,

We appreciate your time and effort in carefully evaluating our manuscript titled “A comparison of two interaction paradigms for training Low Cost Automation assembly in virtual environments”.

We thank you for your suggestion and we improved the future works section adding the provided references as per reviewer request.